# In situ observation of picosecond polaron self-localisation in α-Fe$_2$O$_3$ photoelectrochemical cells

Ernest Pastor [1], Ji-Sang Park [2], Ludmilla Steier [1], Sunghyun Kim[2], Michael Grätzel[3], James R. Durrant[1], Aron Walsh [2,4] & Artem A. Bakulin[1]

Hematite (α-Fe$_2$O$_3$) is the most studied artificial oxygen-evolving photo-anode and yet its efficiency limitations and their origin remain unknown. A sub-picosecond reorganisation of the hematite structure has been proposed as the mechanism which dictates carrier lifetimes, energetics and the ultimate conversion yields. However, the importance of this reorganisation for actual device performance is unclear. Here we report an in situ observation of charge carrier self-localisation in a hematite device, and demonstrate that this process affects recombination losses in photoelectrochemical cells. We apply an ultrafast, device-based optical-control method to resolve the subpicosecond formation of small polarons and estimate their reorganisation energy to be ~0.5 eV. Coherent oscillations in the photocurrent signals indicate that polaron formation may be coupled to specific phonon modes (<100 cm$^{-1}$). Our results bring together spectroscopic and device characterisation approaches to reveal new photophysics of broadly-studied hematite devices.

[1] Centre for Plastic Electronics, Department of Chemistry, Imperial College London, London SW7 2AZ, UK. [2] Department of Materials, Imperial College London, London SW7 2AZ, UK. [3] Ecole Polytechnique Fédérale de Lausanne, Institut des Sciences et Ingénierie Chimiques, Station 6, CH-1015 Lausanne, Switzerland. [4] Deparment of Material Science and Engineering, Yonsei University, Seoul 03722, Korea. Correspondence and requests for materials should be addressed to E.P. (email: e.pastor11@imperial.ac.uk)

Solar energy storage in the form of fuels via a chemical reaction is heavily explored as a means of improving energy security and stability[1,2]. One of the developing technologies is based on photoelectrochemical cells (PECs), which combine a photovoltaic module with a multielectron catalyst to drive photoelectrochemical conversions. In a water-splitting PEC used to make $H_2$, the most demanding reaction is the four electron oxidation of water to oxygen responsible for providing the reducing equivalents to drive the fuel synthesis. In nature, this reaction is achieved within the complex structures of Photosystem II[3], whilst artificially, the best candidates are those based on stable metal oxides. Amongst these, hematite ($\alpha$-$Fe_2O_3$) has received most attention, and is the focus of intensive work in the photocatalytic community[4,5]. This is due to its characteristic red colouration as well as the abundance and intrinsic stability of iron oxides. Despite extensive synthetic efforts to improve PECs based on $\alpha$-$Fe_2O_3$ over the last decades, conversion efficiencies remain modest[4]. Previous research has indicated that ultrafast recombination of electrons and holes is the major efficiency loss channel[6]. However, more recently, a series of ultrafast X-ray studies have suggested that the light-induced reorganisation of the crystal structure in the presence of excess charges, known as polaron formation, ultimately determines the dynamics of the excited state[7,8]. Such polarons not only impact photoconversion yields but have also been postulated to limit the device photovoltage[9], thus restricting charge transfer to the electrolyte. These findings are encouraging as they identify key molecular-level parameters which can be used to aid new synthetic strategies that improve photocatalytic performances. However, while the importance of polaron formation is increasingly accepted, many aspects of the energetics, formation kinetics and recombination dynamics within actual working devices have remained inaccessible by all-optical spectroscopic methods, and are therefore the subject of intense debate[8–11].

A polaron is formed when an injected charge carrier interacts with the surrounding lattice and deforms it. In polarisable, soft materials, a strong electron–phonon coupling can cause the charge to self-trap[12,13]. Depending on the nature of the electronic structure and the extent of the electron–lattice interaction, both highly localised small polarons, or more delocalised large polarons, can be formed[12,14,15]. Polaron formation can impact photocatalysis in several ways. On one hand, the relaxation associated with self-trapping effectively reduces the energy of the photogenerated charge, limiting the photoinduced Fermi level splitting of the material, and thus the photovoltage that can be obtained. On the other hand, the movement of the self-trapped charge is slow and thermally activated, which can lead to higher recombination losses thus limiting the lifetime of reactive charges and consequently photoconversion yields[10,16]. The transport of small polarons is significantly slower than that of large polarons or band transport that is characteristic of conventional semiconductors. For instance, electron transport in $\alpha$-$Fe_2O_3$ is $10^5$–$10^6$ times slower than that of the band-type transport in Si[17]. Such sluggish transport properties have a negative impact on small-polaron conductors used for energy-conversion applications, and in hematite results in the need to shorten the diffusion path to a few nanometres by using meso-structured electrodes[18]. Polarons are particularly relevant to photocatalysts based on transition metals, as these systems are prone to carrier localisation phenomena due to a strong electron correlation within the valence $d$-orbitals[19]. Specifically for $\alpha$-$Fe_2O_3$, recent extreme-ultraviolet (XUV) experiments have demonstrated the formation of localised (electron) $Fe^{2+}$ polaronic states upon photoexcitation[7,20]. These groundbreaking experiments have opened new avenues for exploring the photophysics of $\alpha$-$Fe_2O_3$; however, due to the complex nature of the techniques, the measurements cannot be performed under operation conditions (e.g., low illumination or interactions with electrolyte and fields in an electrochemical cell)[21,22], and thus may not provide device-relevant data. In this work, we directly address this issue and employ a device-based methodology to elucidate the role of small polarons in an $\alpha$-$Fe_2O_3$ photoelectrochemical device.

## Results

**Energetics of polaron formation.** Figure 1a shows a real-space image of a delocalised electron injected into $\alpha$-$Fe_2O_3$. Initially, the

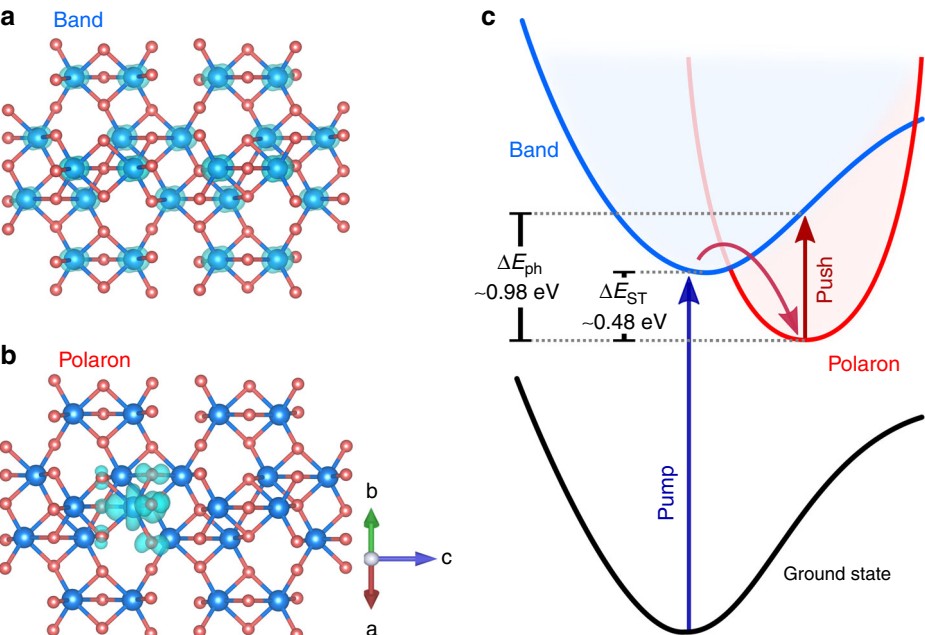

**Fig. 1** Excess charge localisation and delocalisation in $\alpha$-$Fe_2O_3$: spin density associated with an excess electron in $Fe_2O_3$ in (**a**) a delocalised band state spread over equivalent Fe atoms and (**b**) a small-polaron state forming a single Fe(II) species. **c** The potential energy surface for the ground and excited states showing the absence of a barrier for polaron formation (see the Methods section for details)

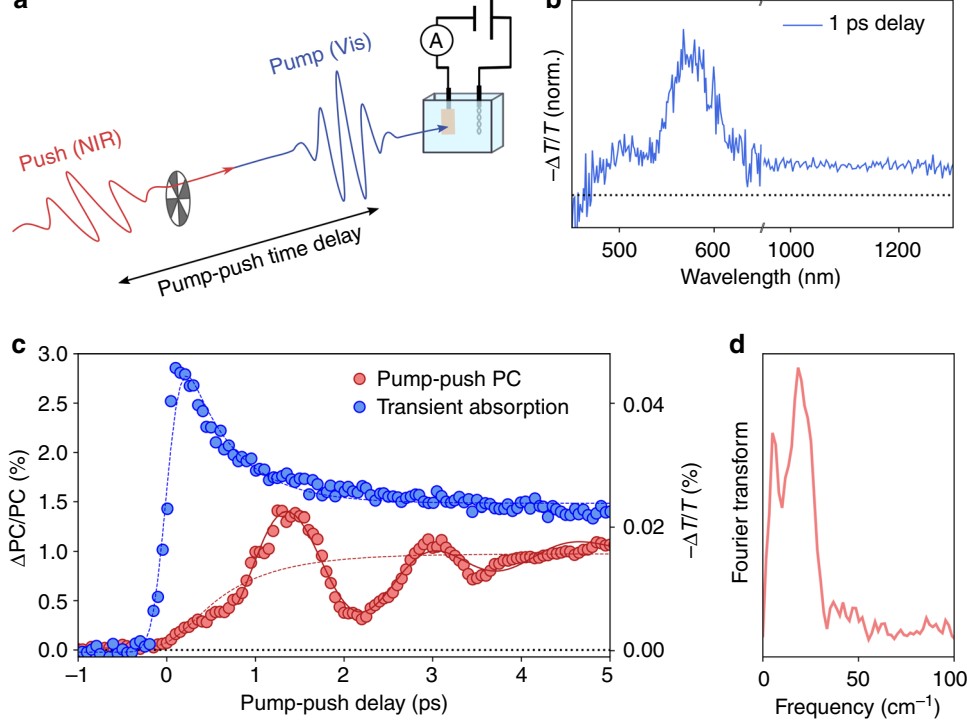

**Fig. 2** Excited-state characteristics and pump–push-photocurrent measurement of an α-Fe$_2$O$_3$ photoanode. **a** Schematic representation of the pump–push-photocurrent detection (PPPC) setup for photoelectrochemical cells (PECs); in this approach, a visible-pump photoexcites the sample, and a push modulates the excited state; the resulting changes in device activity (photocurrent, PC) are monitored as a function of the time delay between pump and push. **b** Transient absorption spectrum of α-Fe$_2$O$_3$ normalised (noted as: 'norm.') in the visible and NIR after photoexcitation with 400 nm light. **c** Decay kinetics of the TA (signal average 1200–1250 nm) and PPPC response at 1200 nm and 1.3 V vs Pt; note PPPC monitors changes in photocurrent (PC). **d** Fourier Transform of the oscillating component of the PPPC data

electron wave function is delocalised across equivalent Fe centres, forming a diffuse band state. The excess electron can become localised on a single site to form Fe (II), with some polarisation of the neighbouring oxide ions (Fig. 1b). This process involves a reorganisation of the lattice (change in configuration coordinate). First-principles calculations show that the stabilisation energy gained upon lattice distortion and localisation, known as the self-trapping energy, is $\Delta E_{ST} = 0.48$ eV, and that the localisation process is barrierless (Fig. 1c). A similar value for the trapping energy is obtained within a continuum dielectric model (see the Methods section for details)[23]. From the self-trapping energy, we estimate an activation barrier for polaron hopping, or transfer between neighbouring localised states, of ~0.2 eV, in agreement with previous reports[17,24,25], and a distortion radius of r = 2.11 Å, shorter than the Fe–Fe interatomic distance (R = 2.99 Å), confirming the small nature of the polaron. In addition, the large trapping energy and the lack of a barrier are consistent with the ultrafast formation of polarons observed using X-ray methodologies[7,20]. We also find that the localised (small-polaron) state can be readily re-excited to a delocalised band stated by photons of just $\Delta E_{ph} = 0.98$ eV (~1265 nm). While this is significantly larger than thermal energy ($k_B T = 0.025$ eV at 300 K), this excitation could be readily provided by near-IR photons in the solar spectrum, effectively transferring the localised electron to a more mobile, delocalised state[26].

In this work, we take advantage of the possibility of re-exciting polarons with IR photons in order to study their formation and operational relevance. We use a device-sensitive technique known as visible-pump-IR-push spectroscopy with photocurrent detection, which has been successfully employed to study polaron states in solid-state organic and hybrid solar cells[27], but has not yet been applied to photocatalytic systems. In this technique, a pump pulse is used to generate an excited state, and a subsequent IR-push-pulse transfers this population to higher-lying levels (Figs. 2a, 1c). This IR transition can direct the electron dynamics along an alternative photophysical pathway and affect the device performance. The resulting changes in device output (i.e., the photocurrent, PC) are recorded as a function of the time delay between the pump and push[28]. To resolve the formation of localised states in time, we studied a high-efficiency photoelectrode based on a thin (<20 nm) hematite layer[29], which facilitates the extraction of excited carriers before they re-localise. Based on our calculations, we expect that if polaron formation takes place on ultrafast timescales, their re-excitation (push) to a more mobile delocalised state, shown in Fig. 1c, should lead to an enhancement of device efficiency. Details of the experimental setup can be found in the Methods section.

**Optical and photocurrent detection of small polarons**. The availability of an IR transition from localised to delocalised states was first verified using transient absorption (TA) spectroscopy. Figure 2b shows the TA spectrum of α-Fe$_2$O$_3$. The spectrum is characterised by a photoinduced absorbance across the visible and NIR regions. Whilst the visible features have been mostly assigned to photogenerated holes[30,31], the flat spectrum at long (NIR) wavelengths has been associated with the absorbance of electrons[32]. This signal in α-Fe$_2$O$_3$ is primarily attributed to Fe$^{2+}$ centres (Fe$^{3+}$ + 1e$^-$)[30,32]. Therefore, the absorption of NIR light (photon energy ~1 eV) brings electrons to a higher-lying state with a more delocalised wave function. As expected, the TA spectrum is structureless at different times making it a complex

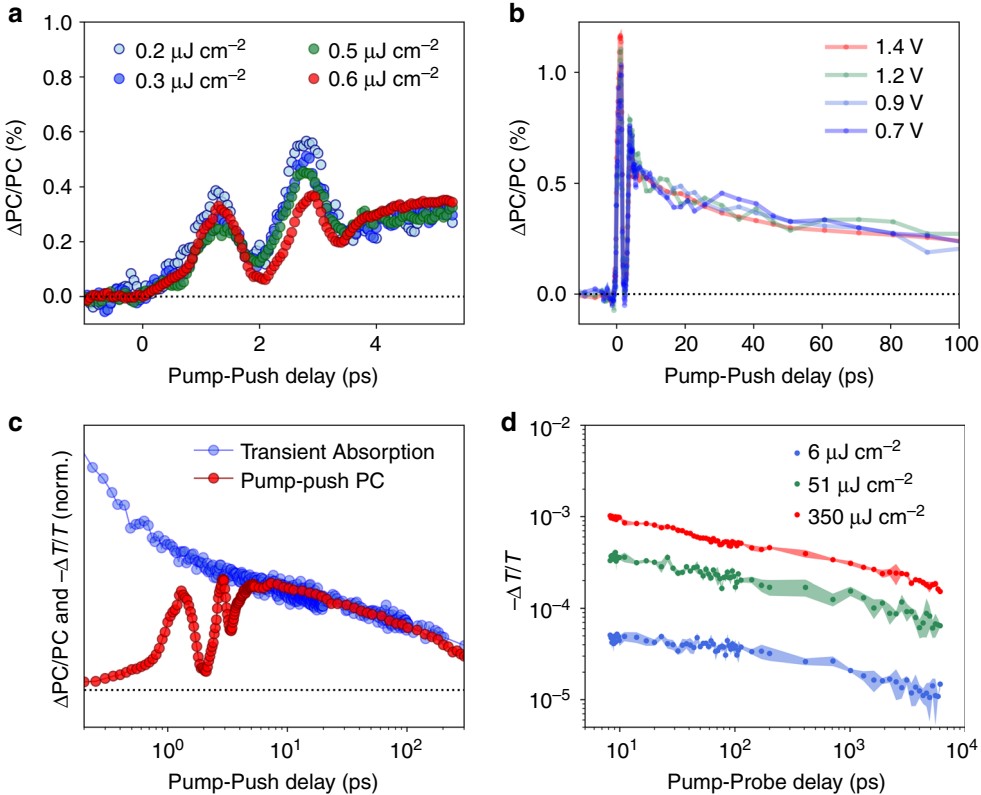

**Fig. 3** Charge carrier density and field dependence. **a** Pump–push-photocurrent (PPPC) response at early times of the α-Fe$_2$O$_3$ PEC at different pump powers and a fixed push power per pulse (1.7 μJ pulse$^{-1}$). **b** PPPC response at a wider time range at different applied potentials vs a Pt counter electrode (see the Methods section for details). **c** Comparison of the TA and PPPC signals across a broad time range. While TA is primarily sensitive to all states in the system (localised and delocalised), the PPPC is only sensitive to bound (localised) states; at early times, when polaron formation occurs, the two signals have different profiles. At longer times, when the polaronic state dominates, both assays show the same response. **d** TA decay at different pump powers (coloured dots and standard deviation) showing a power-law decay kinetics characteristic of recombination via trap/defect states as previously reported for α-Fe$_2$O$_3$ and other metal oxides

assay of dynamics on its own (Supplementary Fig. 1). The NIR TA kinetics, reflecting the population of electronic states, are shown in Fig. 2c (blue dots). When the probe arrives after the pump, we observe a sharp increase in absorbance. The rise time is instrument limited, indicating the interaction of the probe with a population created immediately after excitation. Subsequently, the signal decays sharply with a time constant of ~600 fs and then plateaus, later being followed by a much slower decay. At high-excitation intensities, we observe an additional fast decay component that accelerates with increasing fluence (Supplementary Figs. 2, 3). We attribute this to the non-geminate interaction between excited states at high-excitation densities. Similar kinetics on ultrafast timescales have been observed in other studies[32]. However, in this work, we focus on the intensity-independent component in the decay. This component, with a time constant of ~600 fs (Fig. 2c, Supplementary Fig. 3), is the only component present at the illumination intensities relevant for device operation.

The Fig. 2c (red dots) shows pump–push-photocurrent (PPPC) data from an α-Fe$_2$O$_3$ PEC obtained after pumping at 400 nm and pushing at 1200 nm (0.96 eV). Contrarily to the TA data, we do not observe a sharp rise of the PPPC signal. Instead we observe a gradual increase in the response within the first ~600 fs, which plateaus after ~3 ps to yield a long-lived state (see Fig. 3c for a larger time range). Because PPPC is only sensitive to the bound states (which lead to losses in the device performance), the observed PPPC signals likely reflect the trapping and/or self-

localisation of photogenerated carriers. As expected for resonant interactions, we find that the photocurrent signal amplitude varies linearly with increasing push intensity for a fixed pump power (Supplementary Figs. 4, 5) and with increasing pump power at a fixed push intensity (Supplementary Fig. 6). Interestingly, at early times, the signal can be represented as the convolution of an exponential decay with an oscillatory feature characteristic of the superposition of coherent (likely vibronic) states, as discussed in detail below.

Direct comparison of the TA and PPPC signals in Fig. 2c shows that the decay of the TA component matches the build-up of the photocurrent signal. We propose that photoexcited electrons occupy a delocalised band state, and the transition from this state to higher excited states has a strong transition dipole moment resulting in a strong absorption peak. The formation of the delocalised states is seen as a sharp rise of the TA signal within the time resolution of our measurements. Charges in these states are highly mobile, and do not need extra energy to progress along the photophysical pathway leading to photocurrent generation—this is manifested by the absence of PPPC signal at early timescales. Within ~600 fs of photoexcitation, band electrons localise into a polaronic state with an apparently lower excited-state absorption dipole. This can be seen as the simultaneous decay of the TA signal and the rise of the push-modulated photocurrent. Our photocurrent data are in agreement with the X-ray data in refs. [7,8], which assigned the fast localisation process to the formation of Fe(II) electron polarons. This process

also matches our first-principles calculations, which predict the possibility of re-excitation of small-polaron states to a higher-lying band state with NIR photons. The observed timescale of small-polaron formation is similar to the timescale proposed for $\alpha$-$Fe_2O_3$ and other systems[33], but longer than that reported by Carneiro et al. using XUV spectroscopy[7]. Such discrepancy between experiments could result from differences in excitation power and the density of excited states (lower excitation in this work), or from the differences in the crystallinity of the samples. Nonetheless, the agreement between our device photocurrent data and the X-ray data is striking given the completely different experimental conditions and techniques, and strongly points towards a correlation between the dynamics of the lattice and the electronic energy levels. This is a strong argument for the polaronic behaviour being intrinsic to the material and essential for understanding the device performance.

The oscillatory feature in the PPPC response (Fig. 2c) can be extracted and modelled using a damped sinusoidal function with two frequencies (solid line). Analysis of the isolated oscillation (Fig. 2d, Supplementary Fig. 7) reveals two characteristic frequencies at $6.7\,cm^{-1}$ and at $20\,cm^{-1}$, which was associated to confined acoustic vibrations of the oxide lattice[34]. Such oscillatory behaviours are commonly related with the coherent superposition of vibronic states, and can reveal information about the electron–phonon coupling in soft semiconductors[35]. In our case, this superposition cannot be prepared by short optical pulses, as polaron states are not populated directly. Instead coherent beatings are impulsively generated via a fast charge transfer between two energy surfaces (Fig. 1c), as the energy uncertainty associated with this process ($\sim 55\,cm^{-1}$) is controlled by the transfer rate ($\sim 0.6\,ps$). Since we only observe the oscillations in the PPPC signal, we propose that the observed oscillation frequencies reflect the dynamics of the $\alpha$-$Fe_2O_3$ lattice that are specific to the polaronic state, as addressed by the push. We note that higher frequency phonons might also be coupled to the electronic modes; however, the duration of our laser pulses and particularly the rate of polaron formation ($\sim 600\,fs$) limit the maximum frequencies of any coherent oscillation that we can observe to $\sim 55\,cm^{-1}$. While further work is needed to unambiguously establish the nature of the oscillations and elucidate the mechanism by which vibrational modes couple with polaron states, we hypothesise that direct control of such modes might lead to vibronic tuning of the polaronic state as observed in other soft materials and open new routes for device improvements[36].

Our photocurrent measurements reveal the formation of localised, bound states that are detrimental for device performance. While calculations predict an intrinsic small-polaron state, photocatalytically active semiconductors are complex systems which can have a broad range of defects (e.g., oxygen vacancies or grain boundaries) that could also trap carriers[37,38]. Here, we are primarily concerned with events taking place within 1–2 ps of excitation when various structural changes are expected[39,40]. Previous studies have shown that the TA response of hematite in these timescales is relatively independent on charge-carrier density or the presence of electric fields, and are even comparable between thin films and single crystals[6,32,41]. These observations are an indication that the sub-picosecond photophysics are dominated by intrinsic rather than extrinsic factors. Furthermore, they can also explain the consistencies in the reported experimental findings despite variations in sample preparation or measurement techniques[5].

To further confirm the nature of the localised states we probe in our experiments, we performed measurements as a function of excitation intensity and applied potentials (Fig. 3a, b, respectively). In the former case, we do not observe a strong dependence

of the carrier localisation kinetics on the concentration of carriers in the system. Instead we observe that the yield (number of localised states per extracted carrier) remains constant at different photon densities. Equally, in the latter case, we do not observe a strong field dependence of the decay within the first 100 ps after excitation. Such behaviour is consistent with the formation of small-polaron states that are intrinsic to the material. Comparison of the TA and photocurrent data in Fig. 3c shows that the signals match at longer timescales, while the differences at early times can be explained by the contribution of the delocalised band states to the TA. This indicates that beyond a few picoseconds, the polaronic state governs both the optical and photocurrent responses. In this context, it has previously been reported that the TA characteristics of hematite and other metal oxides at long timescales (ns-s) are sensitive to both the excitation intensity and the applied potential[42–44]. For example, larger anodic applied potentials result in slower kinetics due to the suppression of recombination pathways. This recombination is non-geminate and is therefore found also to accelerate at higher carrier densities. Similar effects were observed for highly correlated oxides, where polaron formation was inferred by X-ray methods, and in which recombination has been proposed to be mediated by defect states[45]. In agreement with these studies, we observe a dependence of the TA signal on excitation power (Fig. 3d, Supplementary Fig. 8) and on the applied potential (Supplementary Fig. 9). This suggests that electron polarons recombine at longer timescales, possibly via trap/defect states, giving rise to characteristic power-law decays, as shown in Fig. 3d.

## Discussion

Polarons play an important role in the charge-carrier mobilities and energetics of organic electronic materials[46]. Similarly, in perovskite solar cells, large polarons have been proposed to prevent recombination and aid performance[14,47]. Despite the relevance of polaronic states in optoelectronic materials, their role in photoelectrochemical devices has hardly been explored. While strong carrier localisation is generally negative in solar cells (hindered charge extraction), its role in solar-fuel devices might be more complex. For example, delocalised majority carriers might favour charge extraction leading to increased efficiencies. Conversely, localisation of minority carriers might enable multi-electron catalysis on atomic sites, thus improving selectivity for the desired transformations. Such a complex balance could explain why $\alpha$-$Fe_2O_3$, despite its polaronic character, remains one of the best photoanode materials. A detailed understanding of the interplay between localisation and delocalisation, and their impact upon carrier dynamics and energetics, is critical to make realistic estimates of photoconversion targets and guide the design of new materials.

In summary, we have tracked the formation of small polarons in $\alpha$-$Fe_2O_3$ in real time and demonstrated their impact on photoelectrochemical systems. By employing, for the first time, an optical-control method with photocurrent detection we have bypassed the limitations of all-optical techniques and exposed the impact of ultrafast lattice relaxations in $\alpha$-$Fe_2O_3$ devices. We expect these phenomena to occur in other metal oxides, especially those based on transition metals. Kinetically, electron polarons impact PECs by limiting transport and charge separation, thereby facilitating the loss of carriers via recombination; this is particularly critical in PECs involving complex and slow catalytic processes. Energetically, the stabilisation of charges by the lattice relaxation (Fig. 1c) can also limit the photovoltage that the PEC can attain and directly limit the chemical transformation that can be achieved[48]. Indeed, this path has recently been postulated as a mechanism to explain the consistently modest solar-to-chemical

conversion efficiencies of hematite photoelectrodes[9]. Based on our calculations and device measurements, we propose that further synthetic developments must target materials that sustain delocalised wave functions of majority carriers. This might be achieved by reducing the lattice reorganisation energy ($\Delta E_{ST}$ in Fig. 1c), thus blocking the formation of the polaronic state, or by increasing the barrier for charge localisation. In addition, the observation of lattice phonon modes of ~7 and ~20 cm$^{-1}$ and their specificity to polaronic states can provide additional information about the dynamics of localisation and how it can be vibronically controlled. From a synthetic viewpoint, modifying the reorganisation energy or localisation barriers is likely to require direct changes in the ligands which bind to the metal centre, or changes in the coordination environment[49]. In this context, strategies like the formation of metal oxo-hydroxides and/or the use of multinary oxides, such as $ZnFe_2O_4$ ferrite materials[50], might aid the control of polaronic states and thus improve device efficiencies for sustainable solar-fuel production.

## Methods

**$Fe_2O_3$ sample preparation.** Hematite samples were prepared by atomic layer deposition (ALD), as previously reported[29]. For this study, hematite photoanodes consisting of a 20 nm thin hematite film on a 1.5 nm ALD $TiO_2$ layer deposited on FTO were used. After ALD, the electrode was annealed at 500 °C for 2 h (10 °C/min ramp rate). For, TA and PPPC measurements, a contact wire was soldered to the FTO contact, and was insulated with epoxy (E9234, Loctite).

**Pump–push photocurrent measurements (PPPC) and transient absorption (TA) setup.** (Figs. 2, 3) A 800 nm 4 kHz Ti:Sapphire regenerative amplifier (Astrella, Coherent) produced ~35 fs pulses that were used to seed an optical parametric amplifier (TOPAS-Prime, Coherent) and β-barium borate (BBO) doubling crystal. BBO produced 400 nm light by second harmonic generation. This 'pump' was then delayed using a mechanical stage. The 1200 nm 'signal' output of the TOPAS, modulated by a mechanical chopper at ~1.2 kHz, was used as the 'push'. The pump and push beams were focused onto a ~0.2-mm$^2$ diameter spot on the electrochemical cell. A lock-in amplifier, synchronised to the modulation frequency of the pump (4 kHz) and push (~1.2 kHz) recorded the photocurrent. $\Delta$PC = push-induced photocurrent; PC = pump-induced photocurrent. Measurements were performed on an electrochemical cell with quartz windows which was filled with 1 M NaOH (pH 13.6). In order to enable push-induced photocurrent detection with low noise levels, a two-electrode cell with Pt as a counter electrode and moderate applied potentials (ranging between 0.7 and 1.4 V vs Pt) were required. A custom-made battery-based power supply was employed to apply a constant voltage. For TA measurements a three-electrode configuration could be used, and a wider applied potential range was explored. As shown in Supplementary Fig. 9, the TA kinetics at short timescales are independent of applied potential and become field dependant at longer time delays.

For the TA measurements, seed pulses (800 nm, <100 fs) were generated at a repetition rate of 1 kHz by a Ti:Sapphire regenerative amplifier (Spectra-Physics Solstice, Newport Corporation), and routed towards an optical parametric amplifier (TOPAS, Light Conversion) coupled to a frequency mixer (NIRUVis, Light Conversion) to provide the 400 nm pump. The seed pulses were directed to a mechanical delay stage, then directed to a commercially available femtosecond transient absorption spectrometer (HELIOS, Ultrafast Systems). Therein, the pump (modulated at 500 Hz) and the broadband near-infra-red (~850–1350 nm) probe were focused onto a ~0.5 mm$^2$ spot on the sample. The bias-dependent measurements reported in Supplementary Fig. 9, used a 420 nm pump (14 μJ cm$^{-2}$, equivalent to 1.3 10$^{13}$ photons absorbed being on the same order of magnitude as the charge-carrier density in these films $N_D = 6.5$ 10$^{18}$ cm$^{-3}$) and a broadband visible (~400–800 nm) probe. The measurements were carried out in an electrochemical cell in a three-electrode configuration using a Ag/AgCl (sat'd KCl) as a reference, a Pt wire as a counter and the hematite photoanode as the working electrode (illuminated from the hematite/electrolyte side). The potential was applied using a PGSTAT101 (Autolab).

**Details of polaron modelling**

*Dielectric polaron model.* The driving force for polaron formation is a balance between the kinetic energy loss and potential energy gain associated with localising an excess electron and polarising the host crystal. In the simplest description, which was further developed by Fröhlich and others in a self-consistent form, the polaron binding energy depends on the electron effective mass ($m^* = 4.6$ m$_e$) and dielectric constants (at high-frequency $\varepsilon_\infty$ and low frequency $\varepsilon_0$).

$$E_P = \frac{1}{4\pi^2}\frac{m^* e^4}{2\hbar^2 \varepsilon_{eff}^2} \tag{1}$$

where the effective dielectric screening ($\varepsilon_{eff} = 5.66$) is given by

$$\frac{1}{\varepsilon_{eff}} = \frac{1}{\varepsilon_\infty} - \frac{1}{\varepsilon_0} \tag{2}$$

The numbers in parentheses are the calculated isotropic averages of the effective mass and dielectric tensors for $Fe_2O_3$, resulting in a predicted value of $E_P = 0.49$ eV for $Fe_2O_3$.

**First-principles electronic polaron model.** The self-trapping energy of a polaron can be calculated directly from the total energy difference between two models containing a delocalised and localised carrier, respectively (Fig. 1). Calculations performed using a first-principles supercell procedure for an excess electron yield $E_{ST} = 0.48$ eV for $Fe_2O_3$. The barrierless nature of the transition was found by performing a series of total energy calculations that map along the configurational coordinate between the pristine (bulk) and the distorted (polaron) structures.

We estimated the activation energy for polaron hopping as $\Delta E_{ST} \sim 2\Delta E_{a,hop}$. Using this approximate formula[25], which assumes a diabatic regime we obtain: $\Delta E_{a,hop} \sim 0.2$ eV.

Here, spin-polarised density functional theory (DFT) calculations were performed using the projector-augmented wave (PAW) method[51] to describe the interaction between ions and electrons, as implemented in the Vienna ab-initio Simulation Package (VASP)[52]. In the calculations, the delocalised state was obtained by adding an electron to the conduction band edge, collectively reducing the magnitude of the magnetic moments of all Fe in the system. The localised polaron state, on the other hand, was formed by changing the magnetic moment of a specific Fe atom. The atomic structure of $Fe_2O_3$ with an additional electron carrier was optimised to define the two local minima of the configuration coordinate. The extent of localisation for an excess carrier can be readily visualised through the self-consistent spin density. We employed a $2 \times 2 \times 1$ supercell (hexagonal setting) containing 120 host atoms and a set of k-points generated by $3 \times 3 \times 2$ mesh for Brillouin zone integration. The wave functions were expanded in plane waves with an energy cut-off of 400 eV. The exchange-correlation functional suggested by Perdew, Burke and Ernzerhof[53] was first used to relax the internal coordinates until the residual force were <0.01 eV/Å. Final production calculations were performed using a non-local hybrid exchange-correlation functional with 25% screened exact exchange as suggested by Heyd, Scuseria and Ernzerhof[54].

## Data availability

The data of this study are available upon request.

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

## Acknowledgements

The authors thank Tom Hopper for his comments on the paper. We are grateful to the UK Materials and Molecular Modelling Hub for computational resources, which is partially funded by EPSRC (EP/P020194/1). A.A.B. is Royal Society Research Fellow. This project has also received funding from the European Research Council (ERC) under the European Union's Horizon 2020 research and innovation programme (Grant Agreement No. 639750). This research was supported by Creative Materials Discovery Program through the National Research Foundation of Korea (NRF) funded by Ministry of Science and ICT (2018M3D1A1058536). L.S. would like to thank the European Research Council (H2020-MSCA-IF-2016 Grant 749231) for funding.

## Author contributions

E.P. and A.A.B. conceived the study. L.S. produced and characterised the samples and carried out voltage bias-dependent ultrafast absorption experiments in the visible region. E.P. planned and performed the ultrafast experiments. A.W., J.P. and S.K. performed theoretical calculations. E.P., A.A.B., L.S., M.G. and J.R.D. discussed the implications of polaron formation for photoelectrochemical device performance. E.P. and A.A.B. wrote the paper with input from all authors.

## Additional information

**Competing interests:** The authors declare no competing interests.

