## [Peer Review File · Nature Communications]

Reviewers' comments:

Reviewer #1 (Remarks to the Author):

This manuscript investigates the mechanism of limited efficiency in hematite Fe₂O₃ photoanodes. Recent spectroscopic results have suggested that small polaron formation limits transport of majority carriers in Fe₂O₃; however, these experiments are generally *ex situ* and focus on the early time kinetics. Consequently, it has remained to be seen to what extent small polaron formation influences actual device efficiency in working photoelectrochemical cells. To address this question, the authors have applied pump-push-photocurrent spectroscopy to investigate the effect of ultrafast small polaron formation on water oxidation efficiency in Fe₂O₃ photoanodes.

It is shown that pumping the self-trapped small polaron state with a NIR push pulse re-excites the polaron to a free-carrier state, leading to enhanced photocurrent yields. This provides strong evidence that electron self-trapping is efficiency limiting in actual Fe₂O₃ devices. Despite very different experimental conditions and measurement techniques, these results show remarkable agreement with kinetics measured by previous spectroscopic methods (approximately 600 fs for polaron formation). This work is impactful in that it presents a strong correlation between ultrafast electron trapping via small polaron formation and PEC efficiency. It also suggests new routes for improving Fe₂O₃ water oxidation efficiency based on this improved understanding.

I am glad to recommend this work for publication. However, below are several questions, which the authors should address prior to publication.

Question 1:

Although I am not an expert on the theory aspect of this paper, I have a few questions about the DFT calculations. First, how were the localized and delocalized states defined and what were the associated constraints for these calculations? Is it correct to assume that charge density was constrained in each state as either evenly distributed across Fe cations or localized to a single Fe cation and then the lattice was allowed to relax in order to define a reaction coordinate?

Question 2:

The DFT calculations show no barrier to polaron formation. The authors note the similarity of this result to experimental reports, specifically Ref. 7 showing fast formation rates. Although qualitatively similar, it is important to note that the barrier to polaron formation based on rates reported in Ref. 7 is actually 190 meV (see also Ref. 49).

Question 3:

The authors report a barrier 0.2 eV for polaron hopping. However, I was unable to determine how this barrier was calculated.

Question 4:

One of the most interesting observations of this paper is coherent coupling of small polaron transport to acoustic phonon modes. Noting that small polarons are typically formed by higher frequency optical phonons, do the authors have any hypotheses about how this coupling occurs? Another way of asking this question is how do acoustic phonons influence polaron transport given that the wavelength of an acoustic phonon is significantly larger than the size of a small polaron?

Question 5:

Lastly, in the concluding paragraph of the paper, the authors address the important question of why trapping of majority carriers (electrons) should have a significant influence on the hole-driven water oxidation reaction (minority carriers). They indicate that electron mobility limits charge separation which determines minority carrier lifetime. This is an important argument, and I simply suggest that this be emphasized earlier in the manuscript to make this point clear.

Other:

In the concluding paragraph the stabilization energy is given the symbol Δ_{GST} , compared to Δ_{EST} , which is used in Figure 1.

Reviewer #2 (Remarks to the Author):

Pastor *et al.* present an ultrafast pump-push-photocurrent study of a photoelectrocatalytic device employing a hematite thin film as the active layer to investigate the dynamics of carrier self-trapping following photoexcitation. Using ~ 1 eV push photons, they are able to drive trapped carriers into mobile, delocalized states, leading to an increase in photocurrent. From these measurements, they find that electron small polaron formation occurs with a ~ 600 fs time constant. They also show that the photocurrent increase oscillates strongly as a function of pump-push delay time for the first few ps and identify two acoustic phonon modes (6.7 and 20 cm^{-1}) responsible for this modulation. Finally, they propose that near-IR solar radiation could be harnessed to drive the same process to boost the carrier mobility (and thereby efficiency) of hematite photoelectrocatalytic devices.

Hematite is a very promising anode material for photoelectrocatalytic water splitting, but much remains unknown about its carrier dynamics at early times. This work is incredibly timely and offers a key insight in this regard by employing a non-optical probe of ultrafast dynamics. Crucially, these experiments are performed under working conditions, and this is the first application of the technique to the study of a photocatalytic material. The data are compelling and well-presented, and the authors provide sufficient experimental detail for other researchers to reproduce their work. This work will likely be of great interest to both the photocatalysis and ultrafast communities. I strongly support publication in *Nature Communications*.

I have two very minor criticisms:

On page 4, the authors state, "Subsequently the signal decays sharply with a time constant of ~ 600 fs and then plateaus, later being followed by a much slower decay. At high excitation intensities, we observe an additional fast decay component that accelerates with increasing fluence." I would expect some quantification of the TA data in the Supplementary Material (plotting of fits, tabulation of time constants and their amplitudes, etc.), but this is not presented.

On pages 7-8, they state, "In our case, this superposition can be prepared by short optical pulses or impulsively generated via a fast charge transfer between two energy surfaces (Figure 1c). Since we only observe the oscillations in the PPC signal, we propose the latter scenario and conclude that the observed oscillation frequencies reflect the dynamics of the α -Fe₂O₃ lattice that are specific to the polaronic state, as addressed by the push." I find this confusing, as the oscillation is a function of pump-push delay time. A coherent vibronic state can only be observed in an ensemble measurement such as this because of the initial short optical pulse, regardless of whether it is in the delocalized or polaronic excited state.

It would also be interesting in a future publication if the authors presented PPC data using push photons of different energies. Assuming the assignment of the NIR TA signal is correct (stronger ESA from the initial band state, weaker absorption from the polaronic state), the oscillation would have to arise from coherent nuclear wavepacket motion on the polaronic potential energy surface, because an oscillatory transfer between the two states would also manifest in the TA trace. This conclusion would be further supported if a stronger modulation were observed at lower push photon energies and a weaker modulation were observed at higher push photon energies.

Dugan Hayes
University of Rhode Island

We thank the reviewers for their constructive feedback, experiment suggestions and the points they have raised. Below we provide answers (in blue) for their questions.

Reviewer #1 (Remarks to the Author): *This manuscript investigates the mechanism of limited efficiency in hematite Fe₂O₃ photoanodes. Recent spectroscopic results have suggested that small polaron formation limits transport of majority carriers in Fe₂O₃; however, these experiments are generally ex situ and focus on the early time kinetics. Consequently, it has remained to be seen to what extent small polaron formation influences actual device efficiency in working photoelectrochemical cells. To address this question, the authors have applied pump-push-photocurrent spectroscopy to investigate the effect of ultrafast small polaron formation on water oxidation efficiency in Fe₂O₃ photoanodes.*

It is shown that pumping the self-trapped small polaron state with a NIR push pulse re-excites the polaron to a free-carrier state, leading to enhanced photocurrent yields. This provides strong evidence that electron self-trapping is efficiency limiting in actual Fe₂O₃ devices. Despite very different experimental conditions and measurement techniques, these results show remarkable agreement with kinetics measured by previous spectroscopic methods (approximately 600 fs for polaron formation). This work is impactful in that it presents a strong correlation between ultrafast electron trapping via small polaron formation and PEC efficiency. It also suggests new routes for improving Fe₂O₃ water oxidation efficiency based on this improved understanding.

I am glad to recommend this work for publication. However, below are several questions, which the authors should address prior to publication.

Question 1: *Although I am not an expert on the theory aspect of this paper, I have a few questions about the DFT calculations. First, how were the localized and delocalized states defined and what were the associated constraints for these calculations? Is it correct to assume that charge density was constrained in each state as either evenly distributed across Fe cations or localized to a single Fe cation and then the lattice was allowed to relax in order to define a reaction coordinate?*

The reviewer's description is very close to the approach that we adopted. If we add one electron to the system, then we can imagine two extreme solutions. In one case, one electron is localised into a Fe site and forming an unpaired spin at a narrow site. In the other case, charges are not localised into a specific atom and delocalised across the system. The former is a small polaron, and the latter can be regarded as an electron occupying a conduction band edge state. In the actual calculations, we initially provide the magnetic moment of each Fe atom and subsequently relaxed the structures self-consistently. We

obtained the self-trapping energy directly as the total energy difference between the two extreme scenarios.

To clarify this point we added the following statements to the manuscript in Section 2.2:

“In the calculations, the delocalised state was obtained by adding an electron to the conduction band edge, collectively reducing the magnitude of the magnetic moments of all Fe in the system. The localised polaron state, on the other hand, was formed by changing the magnetic moment of a specific Fe atom. The atomic structure of Fe₂O₃ with an additional electron carrier was optimised to define the two local minima of the configuration coordinate.”

Question 2: *The DFT calculations show no barrier to polaron formation. The authors note the similarity of this result to experimental reports, specifically Ref. 7 showing fast formation rates. Although qualitatively similar, it is important to note that the barrier to polaron formation based on rates reported in Ref. 7 is actually 190 meV (see also Ref. 49).*

The reviewer is right in pointing out the importance of identifying the barrier for polaron formation and polaron hopping as this is critical, especially when comparing materials. In our calculations we obtained the barrier for polaron formation by performing total energy calculations to map along the configuration coordinate between the bulk and polaron structures thus directly generating images between the localised state and the delocalised state. Using this method, we find that the process is barrierless and in the text we point out this, alongside the large self-trapping energy, is in good agreement with the fast formation rates reported in the literature, such as the ones in reference 7 (Nat. Mater. 16, 819–825, 2017).

The kinetic model in Ref. 49 (*J. Phys. Chem. Lett.* 2018, 9, 4120), links the polaron formation rate and the energy barrier for polaron formation. However, we interpret the value of 190 meV not as the energy barrier for polaron formation from a delocalised to localised state, but instead as the barrier for polaron hopping ($\Delta E_{a,hop}$) between localised states. This value is defined in *J. Chem. Phys.* 2005, 122,144305 as the activation energy for nearest-neighbour charge transfer and is in agreement with our approximate estimate of ~ 0.2 eV hopping barrier and other experimental results. We have emphasised these points in the revised version of the manuscript, page 3, line 23, as:

“From the self-trapping energy we estimate an activation barrier for polaron hopping, or transfer between neighbouring localised states, of ~ 0.2 eV, in agreement with previous reports,^{17,24,25} and a distortion radius of $r = 2.11$ Å, smaller than the Fe-Fe interatomic distance ($R=2.99$ Å), confirming the small nature of the polaron. In addition, the large trapping energy and the lack of a barrier are consistent with the ultrafast formation of polarons observed using X-ray methodologies.^{7,20}”

Question 3: *The authors report a barrier 0.2 eV for polaron hopping. However, I was unable to determine how this barrier was calculated.*

We obtained $\Delta E_{a,hop}$ from the self-trapping energy (ΔE_{ST}) as $\Delta E_{ST} \sim 2 * \Delta E_{a,hop}$, which is an approximation assuming a diabatic regime described in Ref. 25 (*J. Chem. Phys.* 2003,118,6425) . We have clarified the approximation in the supporting information, page 3 as:

*“We estimated the activation energy for polaron hopping as $\Delta E_{ST} \sim 2 * \Delta E_{a,hop}$. Using this approximate formula [*J. Chem. Phys.* 118, 6455–6466,2003] which assumes a diabatic regime we obtain: $\Delta E_{a,hop} \sim 0.2$ eV.”*

Question 4: *One of the most interesting observations of this paper is coherent coupling of small polaron transport to acoustic phonon modes. Noting that small polarons are typically formed by higher frequency optical phonons, do the authors have any hypotheses about how this coupling occurs? Another way of asking this question is how do acoustic phonons influence polaron transport given that the wavelength of an acoustic phonon is significantly larger than the size of a small polaron?*

This is a very interesting point indeed. We note that, for creating superposition of states and coherent beatings, an “impulsive” excitation is important. Basically, inversed excitation time provides the frequency bandwidth within which the vibrational modes can be excited. In our experiment, the polaron state is populated on a ~ 0.6 ps timescale, which corresponds to the frequency range of coherent beatings 0-55 cm^{-1} . Therefore, higher frequency optical phonons, even if they are probably largely involved in polaron formation cannot be observed through the coherent beatings in our experiment as the corresponding superposition of states cannot be formed. On the other hand, acoustic phonon modes can be coupled to optical phonons which provides a pathway for observing them in optical experiments; however, further work is needed to establish the precise mechanism. We added the comment on this to the revised text, page 7, line 27, as:

“The oscillatory feature in the PPPC response (Figure 2c) can be extracted and modeled using a damped sinusoidal function with two frequencies (solid line). Analysis of the isolated oscillation (Figure 2d and Supplementary Fig. 7) reveals two characteristic frequencies at 6.7 cm^{-1} and at 20 cm^{-1} which was associated to confined acoustic vibrations of the oxide lattice.³⁴ Such oscillatory behaviors are commonly related with the coherent superposition of vibronic states, and can reveal information about the electron-phonon coupling in soft semiconductors.³⁵ In our case, this superposition cannot be prepared by short optical pulses as polaron states are not populated directly. Instead coherent beatings are impulsively generated via a fast charge transfer between two energy surfaces (Figure 1c) as the energy uncertainty associated with this process ($\sim 55 \text{ cm}^{-1}$) is controlled by the transfer rate (~ 0.6 ps). Since we only observe the oscillations in the PPPC signal, we propose that the observed oscillation frequencies reflect the dynamics of the $\alpha\text{-Fe}_2\text{O}_3$ lattice that are specific to the polaronic state, as

addressed by the push. We note that higher frequency phonons might also be coupled to the electronic modes, however, the duration of our laser pulses and particularly the rate polaron formation (~600 fs) limit the maximum frequencies of any coherent oscillation that we can observe to ~55 cm⁻¹. While further work is needed to unambiguously establish the nature of the oscillations and elucidate the mechanism by which vibrational modes couple with polaron states, we hypothesise that direct control of such modes might lead to vibronic tuning of the polaronic state as observed in other soft materials and open new routes for device improvements.³⁶

Question 5: *Lastly, in the concluding paragraph of the paper, the authors address the important question of why trapping of majority carriers (electrons) should have a significant influence on the hole-driven water oxidation reaction (minority carriers). They indicate that electron mobility limits charge separation which determines minority carrier lifetime. This is an important argument, and I simply suggest that this be emphasized earlier in the manuscript to make this point clear.*

In paragraph #2 (page 2, line 28) we discuss the impact of slow polaron transport and we have modified to following sentence to emphasise the effect on carrier lifetime:

‘On the other hand, the movement of the self-trapped charge is slow and thermally activated, which can lead to higher recombination losses thus limiting the lifetime of reactive charges and consequently photoconversion yields.^{10,16}

Other: *In the concluding paragraph the stabilization energy is given the symbol ΔGST , compared to ΔEST , which is used in Figure 1.*

Thank you for pointing this out. We have amended it.

Reviewer #2 (Remarks to the Author): *Pastor et al. present an ultrafast pump-push-photocurrent study of a photoelectrocatalytic device employing a hematite thin film as the active layer to investigate the dynamics of carrier self-trapping following photoexcitation. Using ~1 eV push photons, they are able to drive trapped carriers into mobile, delocalized states, leading to an increase in photocurrent. From these measurements, they find that electron small polaron formation occurs with a ~600 fs time constant. They also show that the photocurrent increase oscillates strongly as a function of pump-push delay time for the first few ps and identify two acoustic phonon modes (6.7 and 20 cm⁻¹) responsible for*

this modulation. Finally, they propose that near-IR solar radiation could be harnessed to drive the same process to boost the carrier mobility (and thereby efficiency) of hematite photoelectrocatalytic devices.

Hematite is a very promising anode material for photoelectrocatalytic water splitting, but much remains unknown about its carrier dynamics at early times. This work is incredibly timely and offers a key insight in this regard by employing a non-optical probe of ultrafast dynamics. Crucially, these experiments are performed under working conditions, and this is the first application of the technique to the study of a photocatalytic material. The data are compelling and well-presented, and the authors provide sufficient experimental detail for other researchers to reproduce their work. This work will likely be of great interest to both the photocatalysis and ultrafast communities. I strongly support publication in Nature Communications. I have two very minor criticisms:

On page 4, the authors state, “Subsequently the signal decays sharply with a time constant of ~600 fs and then plateaus, later being followed by a much slower decay. At high excitation intensities, we observe an additional fast decay component that accelerates with increasing fluence.” I would expect some quantification of the TA data in the Supplementary Material (plotting of fits, tabulation of time constants and their amplitudes, etc.), but this is not presented.

We have modified Supporting Information Figure 3 (page 6) to include the fittings to the raw data, the time constants and errors obtained as well as an explanation of the model employed.

Intensity (mJcm ⁻²)	Time constant (ps)
0.35	0.308±0.014
0.17	0.406±0.019
0.05	0.43±0.04
0.006	0.54±0.07

On pages 7-8, they state, “In our case, this superposition can be prepared by short optical pulses or impulsively generated via a fast charge transfer between two energy surfaces (Figure 1c). Since we only observe the oscillations in the PPPC signal, we propose the latter scenario and conclude that the observed oscillation frequencies reflect the dynamics of the α -Fe₂O₃ lattice that are specific to the polaronic state, as addressed by the push.” I find this confusing, as the oscillation is a function of pump-push delay time. A coherent vibronic state can only be observed in an ensemble measurement such as this because of the initial short optical pulse, regardless of whether it is in the delocalized or polaronic excited state.

We fully agree with the reviewer fast population is required to prepare superposition of states, but we respectfully disagree that ‘impulsive’ excitation can be only achieved by optical pulse directly. Whenever a population of states is mediated by a ‘fast’ process like charge transfer or charge relaxation, the rate of the process will imply the energy uncertainty and potentially formation of superposition of states. For the process which is ~ 0.6 ps fast, energy uncertainty is ~ 55 cm⁻¹ so the modes spaced by this energy or smaller can be brought into superposition of states. We added a remark on this to the revised manuscript page 7 line 31:

“In our case, this superposition cannot be prepared by short optical pulses as polaron states are not populated directly. Instead coherent beatings are impulsively generated via a fast charge transfer between two energy surfaces (Figure 1c) as the energy uncertainty associated with this process (~ 55 cm⁻¹) is controlled by the transfer rate (~ 0.6 ps).”

It would also be interesting in a future publication if the authors presented PPPC data using push photons of different energies. Assuming the assignment of the NIR TA signal is correct (stronger ESA from the initial band state, weaker absorption from the polaronic state), the oscillation would have to arise from coherent nuclear wavepacket motion on the polaronic potential energy surface, because an oscillatory transfer between the two states would also manifest in the TA trace. This conclusion would be further supported if a stronger modulation were observed at lower push photon energies and a weaker modulation were observed at higher push photon energies.

Dugan Hayes

University of Rhode Island

We thank the reviewer for this suggestion and indeed we are very interested in collecting data at different push energies and are currently re-designing the experimental set-up to be able to achieve this.

REVIEWERS' COMMENTS:

Reviewer #1 (Remarks to the Author):

The authors have carefully addressed the questions raised, and I am glad to recommend this work for publication in Nature Communications.

-Robert Baker

Reviewer #2 (Remarks to the Author):

The authors have largely addressed my primary concern by adding plots of the fits to the TA data referenced in the main text to the Supplementary Material. They have also provided the ~ 600 fs time constant obtained from the fits for each pump fluence. However, they still do not provide any quantitative information regarding the additional ultrafast component observed at high fluences, nor do they provide fit values (i.e. beat frequencies, dephasing times, and rise time) for the fit shown in Fig. 2c in the main text. I do believe these should be included for the sake of transparency and completeness, as they may prove useful to readers in the future. Nevertheless, they do not have direct bearing on the main conclusions of the paper, so I do not think it is absolutely necessary to add. I continue to believe that this manuscript presents very important and timely work with impressive data that supports the authors' conclusions, and I enthusiastically support publication.

Dugan Hayes

Reviewer #1 (Remarks to the Author): *The authors have carefully addressed the questions raised, and I am glad to recommend this work for publication in Nature Communications.*

-Robert Baker

Reviewer #2 (Remarks to the Author): *The authors have largely addressed my primary concern by adding plots of the fits to the TA data referenced in the main text to the Supplementary Material. They have also provided the ~600 fs time constant obtained from the fits for each pump fluence. However, they still do not provide any quantitative information regarding the additional ultrafast component observed at high fluences, nor do they provide fit values (i.e. beat frequencies, dephasing times, and rise time) for the fit shown in Fig. 2c in the main text. I do believe these should be included for the sake of transparency and completeness, as they may prove useful to readers in the future. Nevertheless, they do not have direct bearing on the main conclusions of the paper, so I do not think it is absolutely necessary to add. I continue to believe that this manuscript presents very important and timely work with impressive data that supports the authors' conclusions, and I enthusiastically support publication.*

Dugan Hayes

Following the reviewer's suggestion we have added further fit details to Supplementary Figure 3.

We thank the reviewers for their constructive feedback and detailed suggestions to improve the quality of the manuscript.